# Microbial quality assessment of Niger seed (*Guizotia abyssinica* (Linnaeus f.) Cassini) oil in Gondar city: A laboratory-based cross-sectional study

**Lamrot Yohannes**[ORCID]*, **Tsegaye Adane Birhan**[ORCID], **Mastewal Endalew**, **Jember Azanaw**, **Fasika Weldegebrel**

Department of Environmental and Occupational Health and Safety, Institute of Public Health, College of Medicine and Other Health Sciences, University of Gondar, Gondar, Ethiopia

* lamrotyohannes@gmail.com

## Abstract

Foodborne diseases pose a significant public health challenge worldwide. The increasing availability of edible oils in the market, combined with Ethiopia's lack of stringent quality control and regulatory oversight, raises concerns about their safety. This inadequacy in regulation may contribute to microbial contamination, leading to potential public health risks. A laboratory-based cross-sectional study was conducted from May to July 2021. Twelve samples of Niger seed oil (*Guizotia abyssinica* (Linnaeus f.) Cassini) were collected using a universal sampling technique. In the microbiology laboratory, the samples were aseptically examined for bacterial and fungal contamination using standard microbiological methods and procedures. The collected data were entered and analyzed using Stata Version 14. Mean values and standard deviations were computed, and the results were presented in text and tables. Microbial analysis of the 12 Niger seed oil (*Guizotia abyssinica* (Linnaeus f.) Cassini) samples revealed varying levels of bacterial, mold, and coliform contamination. The identified bacterial species included *Staphylococcus aureus*, *Klebsiella pneumoniae*, and *Pseudomonas aeruginosa*, while *Aspergillus niger*, *Aspergillus flavus*, and *Aspergillus fumigatus* were the predominant fungal isolates. The total aerobic bacterial count ranged from $1.3 \times 10^3$ to $9.2 \times 10^4$ cfu/ml, with the highest recorded mold count reaching $4 \times 10^5$ cfu/ml. Additionally, both total and fecal coliform isolates were detected in the samples. The presence of these microorganisms suggests that the oil processing, production, handling, and storage systems lack proper hygienic handling practices. This finding highlights the urgent need for stringent hygiene measures, enhanced quality control protocols, and strict adherence to food safety regulations throughout the production and distribution processes.

**Data availability statement:** The dataset is freely available in the paper for other researchers.

**Funding:** The author(s) received no specific funding for this work.

**Competing interests:** The authors have declared that no competing interests exist.

**Abbreviations:** APC, Aerobic Plate Count; CFU, Coliform Units; EFDA, Ethiopian Food and Drug Administration; FAO, Food and Agriculture Organization; FC, Fecal Coliform; HACCP, Hygiene and Critical Control Point; MAC, MacConkey; ML, Mililiter; MSA, Mannitol Salt Agar; NAFDAC, National Agent for Food and Drug Administration and Control; PCA, Plate Count Agar; PDA, Potato Dextrose Agar; QMS, Quality Management System; TC, Total Coliform; TSI, Triple Sugar Iron; WHO, World Health Organization.

## Background

Foodborne illnesses pose a significant global public health concern. Vegetable oils are widely consumed by billions of individuals across all age groups. Ideally, edible vegetable oil should be free from harmful microbes, toxic substances, and suspended particles [1,2]. Niger seed *(Guizotia abyssinica (Linnaeus f.) Cassini)*, also known as noug, is an oilseed crop native to Ethiopia. It accounts for slightly over 25% of the country's total oilseed production, primarily cultivated in the highlands of Oromia and Amhara. These regions play a crucial role in global Niger seed *(Guizotia abyssinica (Linnaeus f.) Cassini)* production, with Ethiopia, India, and Myanmar being the leading producers [3–5].

Niger seed *(Guizotia abyssinica (Linnaeus f.) Cassini)* is rich in edible oil (40%), protein (20%), and various fatty acids, including linoleic acid (75%–80%), palmitic and stearic acids (7%–8%), and oleic acid (5%–8%) [6,7]. Studies suggest that consuming dietary fats and oils high in linoleic acid may help prevent cardiovascular conditions [8]. Its applications range from culinary uses to medicinal remedies for skin conditions, as well as green manure for soil enhancement. Additionally, it is utilized in the production of soap, paints, lubricants, and illuminants [9–11].

Seed-borne diseases, such as *Aspergillus Niger*, *A. flavus*, *Penicillium* sp., *Alternaria alternata*, *Rhizoctonia solani*, and *R. bataticola*, have posed significant challenges in India. Similarly, in Ethiopia, *Orobanche minor* and *Dodder* have emerged as serious threats in Niger-producing provinces such as Gojam, Gondar, Shoa, and Wellega [4,12,13]. In many developing countries, including Ethiopia, elevated bacterial and fungal contamination levels in vegetable oils are a prevalent concern. This issue is primarily attributed to unsanitary and inadequate hygiene practices during oil ingredient preparation, production, and storage [14]. Such microbial contamination has the potential to cause various illnesses, some of which may lead to severe health complications or even fatalities [15,16].

The edible oil industry operates in various stages, such as plant planting, seed storage, transportation, production, processing, oil storage, and transportation. Security issues at any stage, such as theft, contamination, or improper handling, can adversely affect the quality of the oil product [17]. There are primarily two methods used for extracting oil from seeds on an industrial scale: solvent extraction, which utilizes n-hexane or petroleum ether as the extraction solvent, with a higher temperature affecting the oil's quality [18], and mechanical oil extraction, which results in a lower oil yield and operates at mild temperatures, ensuring process safety and maintaining product quality [19]. In Ethiopia, the food processing industries are currently facing challenges related to the quality of their products [20–22].

In Ethiopia, the projected total production of Niger seed for the marketing year 2020/21 was expected to reach 705,000 metric tons [1]. At the national level, the majority of Niger seed oil available on the market is obtained through mechanical extraction and an informal, small-scale marketing system. Oil produced at smallholder farms is marketed without quality control measures, and hygienic control of oils is not usually conducted on a routine basis. However, this informal marketing system poses a challenge to ensuring oil quality control in urban and non-urban areas [21]. Therefore, in Ethiopia, the production and consumption of Niger seed oils often take place under

unsatisfactory conditions. To protect consumers from unhygienic oil consumption, addition of extraneous substance and exposure to pathogenic organisms such kind of study is very crucial. However, no study, to our knowledge, has examined the microbial quality of Niger seed oils. Limited information exists about Niger seed oil's physicochemical quality [23–25]. Our primary objective was to evaluate the microbial quality of Niger seed oil *(Guizotia abyssinica (Linnaeus f.) Cassini*, focusing on (1) bacteriological and (2) fungal contamination. Additionally, we investigated coliforms to provide valuable insights into the oil's quality. A deeper comprehension of the microbial quality of edible oils will facilitate intervention and quality control efforts. Furthermore, it will incentivize food processing industries, regulatory bodies, and stakeholders to prioritize the enhancement of edible oil quality from the outset.

## Materials and methods

### Study design and period

From May to July 2021, a laboratory-based cross-sectional study design was carried out.

### Study area

The study was conducted in Gondar City, where the edible oil market comprises a significant number of wholesalers and producers. According to data obtained from the Gondar City Trade and Industry Bureau, there are 76 wholesalers of edible oil in the city. Among them, 22 wholesalers supply both imported and locally produced vegetable oil, 43 deal exclusively in locally produced oil, and 11 operate as both edible oil producers and sellers, serving the population of Gondar and its surrounding areas (Fig 1).

### Sample source and study samples

The study's sample source was all Niger seed oils *(Guizotia abyssinica (Linnaeus f.) Cassini)* available in the market in Gondar City, and the study samples were obtained from oils registered with the Gondar Trade and Industry Bureau and the Ethiopian Food and Drug Administration Authority.

### Inclusion and exclusion criteria

This study comprised Niger seed oils that were licensed and registered by the Ethiopian Food and Drug Administration Authority (EFDA) and the Gondar Trade and Industry Bureau and that were sold in the Gondar market. Excluded from consideration were oils with poorly readable labels, missing manufacturing and expiration dates, and improper labeling.

### Sample size and sampling technique

The current study used a universal sampling technique, which included all 12 brands of Niger seed *(Guizotia abyssinica (Linnaeus f.) Cassini)* oil samples that were available in the city's market. The eight locations from which the samples were acquired were Kebele (smallest administrative unit, essentially a neighborhood or village) 7, 8, 9, 18, 11, 10, 15, and 20.

### Materials

A bio-safety hood, an autoclave, an incubator, a refrigerator (RT34M3723S8/HL/2017 model), and a microscope (CX21FS1 model) were used during the microbiological analysis.

### Sampling procedure

Twelve Niger seed oil samples were aseptically collected by three Environmental Health professionals from eight locations in Gondar City using 250 ml Erlenmeyer flasks and 250 ml of pre-packaged edible oil. Care was taken to prevent air exposure and contamination of the flasks before and during sample collection. The samples were then promptly transported to

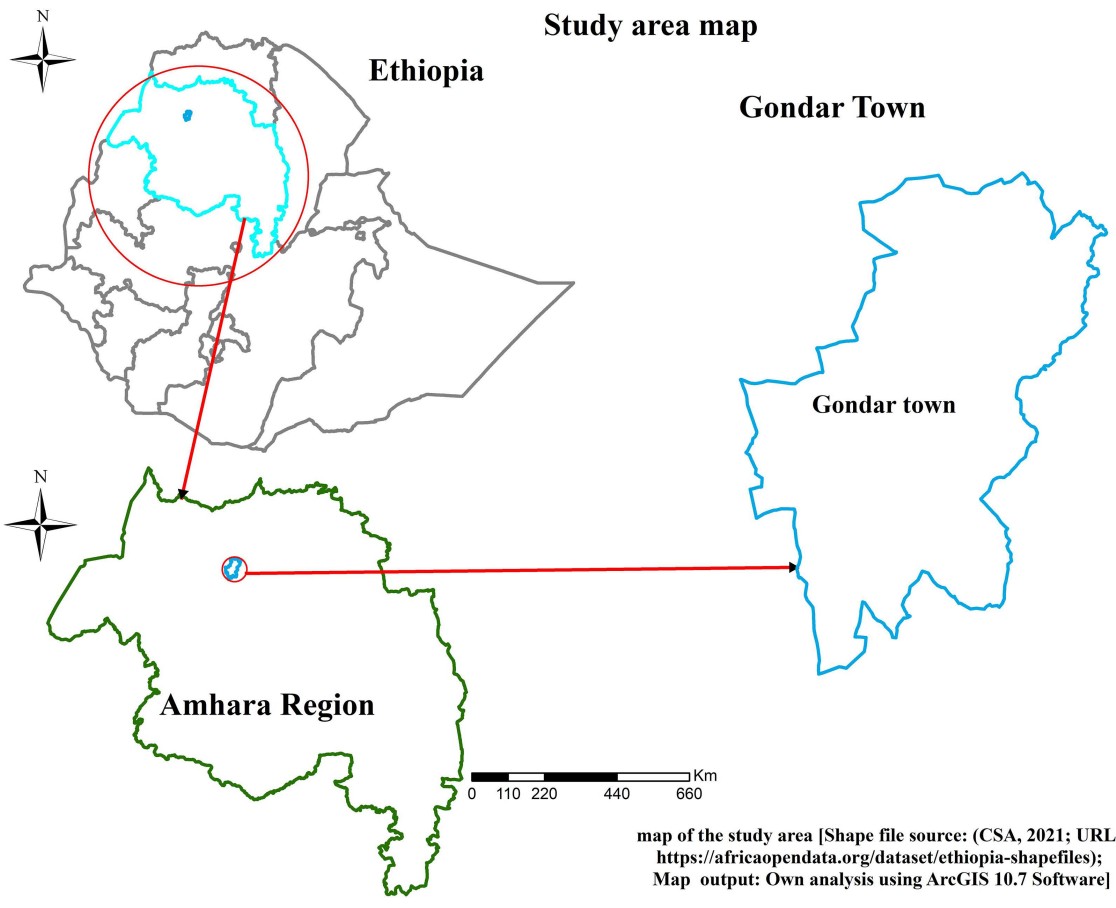

**Study area map**

**Ethiopia**

**Gondar Town**

Gondar town

Amhara Region

map of the study area [Shape file source: (CSA, 2021; URL:
https://africaopendata.org/dataset/ethiopia-shapefiles);
Map output: Own analysis using ArcGIS 10.7 Software]

**Fig 1. Study area map showing the location of Gondar Town.**

the laboratory in an icebox maintained at 4°C. Upon arrival, they were stored in a refrigerator until microbial analysis was performed. Polysorbate 20 was added to enhance emulsification during sample suspension preparation, and all samples were analyzed within 24 hours of collection [26].

## Microbial analysis of oil samples

For the analysis, five distinct growth media were prepared. Nutrient agar, potato dextrose agar (for molds), mannitol salt agar (for *Staphylococcus aureus*), and MacConkey agar (for bacterial isolation) were each dissolved in one liter of distilled water. Additionally, membrane lauryl sulfate broth was prepared for the isolation of total and fecal coliform bacteria. Each medium was sterilized via autoclaving at 121°C and 15 pounds per square inch (psi) of pressure for 15 min.

All essential laboratory equipment, including Petri dishes and volumetric flasks, underwent thorough sterilization. To facilitate the growth of bacteria, coliforms, and molds, buffered peptone water was prepared for the serial dilution process. The incubator temperature was carefully adjusted to a range of 28°C to 37°C to support optimal microbial growth.

## Aerobic plate count

Aerobic Plate Count/APC/ at 30°C/48 hours: Approximately 25 milliliters of oil samples were mixed with 225 milliliters of sterile buffered peptone water. Five- to 6-fold serial dilutions were prepared. Plate Count Agar (PCA) was inoculated with

1 ml aliquots of each dilution using the pour plate method. The plates were incubated at 30°C for 72 hours. After incubation, plates containing 25–300 colonies were selected for colony counting. The results were expressed as colony-forming units (cfu) per milliliter [27].

### Isolation of bacteria

From each sample, 1 ml of edible oil was transferred into 9 milliliters of sterile saline solution and thoroughly homogenized. The homogenized samples were then serially diluted up to six times. After dilution, 100 µl (0.1 ml) of the serially diluted oil samples were plated onto pre-prepared MacConkey Agar (MAC) and Mannitol Salt Agar (MSA) plates. The plates were incubated at 37°C for 24–48 hours. Each dilution was plated in duplicate. After incubation, plates containing 25–250 colonies were selected for colony counting. The results were reported as colony-forming units (cfu) per milliliter [28].

### Isolation of fungi

Each edible oil sample (1 ml, ml) was diluted in 9 milliliters of 0.9% sterile saline solution and thoroughly homogenized. After homogenization, the samples were serially diluted up to six times and plated in duplicate on potato dextrose agar (PDA) using 100 µl (0.1 ml) aliquots. The plates were incubated at 28°C for five to seven days. Each dilution was plated in duplicate. After incubation, colonies ranging from 10 to 35 were enumerated, and the results were expressed as colony-forming units (cfu) per milliliter [27].

### Enumeration, isolation, and identification of total and fecal coliforms

For the analysis of total and fecal coliforms, 1 ml (ml) of each oil sample was diluted in 100 milliliters of sterilized distilled water and filtered through a Millipore membrane filter (45 µm) using a vacuum pump. A membrane lauryl sulfate medium was placed inside an absorbent pad. The membrane-filtered samples were then placed on the membrane lauryl sulfate medium in an aluminum Petri dish. The total and fecal coliform counts were determined by incubating the plates at 37°C for total coliforms and 44°C for fecal coliforms for 18–24 hours. After incubation, yellow colonies were counted as coliforms in both cases. The microbial load (cfu/ml) was calculated using the following formula [29]:

N (for bacteria and fungi) = number of colonies/volumes of oil sample added on a plate*dilution factor*number of plates

$$Fecal\ Coliform = (number\ of\ colonies/volumes\ filtered) * 100$$

$$Total\ Coli\ form = (number\ of\ fluorescent\ colonies + number\ of\ blue\ colonies\ and\ non–fluorescent\ colonies)/volume\ of\ sam$$

Microbial identification was conducted using standard methods described in the medical laboratory manual, microbiological standards, and basic medical microbiology literature [30,31]. The types of bacteria, coliforms, and fungal species were identified based on established protocols.

### Identification of isolates

**Cultural (macroscopic) characterization of bacterial and fungal isolates.** Five isolates were obtained from each young culture, and 0.1 milliliters (ml) of each isolate was extracted. The extracted samples were then spread onto a nutrient agar medium and incubated at 37°C for three days. After the incubation period, the bacterial isolates were identified based on colony morphology, including variations in size, color, and form. The observations were systematically recorded. For macroscopic identification, a standard mycological identification method was employed, considering the appearance and cultural characteristics of the bacterial and fungal isolates (H. L. J. I. g. o. i. f. Barnett, 1960; [32]).

### Microscopic characterization for bacterial and fungal isolates

**Gram stain for bacteria.** Five bacterial isolates from a young culture were heat-fixed on slides and subjected to Gram staining. Cells were stained with crystal violet for 1 min, rinsed, treated with Gram's iodine for 1 min, rinsed twice (a gentle water spray, then 30 seconds in alcohol), and counterstained with safranin for 1 min. After blotting dry, slides were examined under a 1000x oil-immersion microscope (Olympus CX21FS1). Bacterial morphology and Gram reaction guided identification, supplemented by biochemical characterization per Bergey's Manual [32].

### Microscopic identification of fungi using lactophenol blue staining

A fungal culture sample was mounted on a clean slide, stained with lactophenol blue, and covered with a coverslip. Microscopic examination was performed under a light microscope. Identification was based on hyphal and conidiophore features, referencing Enemuor and colleagues [33] and Barnett and Hunter [34].

### Biochemical characterization of bacterial isolates

After Gram staining, the isolated bacteria underwent several biochemical tests, including the triple sugar iron (TSI) test, citrate utilization test, urease test, motility test, and carbohydrate fermentation test (starch hydrolysis test). These tests were conducted following the protocols described by [35,36].

### Data processing, management, and analysis

The raw data was coded and transferred to Microsoft Excel 2016, then exported to STATA for further analysis. To ensure accuracy, consistency, and completeness, data cleaning was performed to identify and correct any missing values or variables. The mean and standard deviation were computed, and results were presented using text and tables.

### Ethics approval and consent to participate

The faculty's Institutional Review Board examined and approved this work. Ethical clearance was obtained from the institutional review board of the Institute of Public Health, College of Medicine and Health Sciences, University of Gondar. A permission letter was also obtained from the Gondar Trade and Industry office. Written informed consent was taken after explaining the purpose of the study. Confidentiality of the information was maintained thoroughly by excluding names as identification in the data and keeping their privacy during data collection, and also individual results were kept secure.

## Result

### Enumerative value of microbes in the oil

In the current study, the highest enumeration value on MSA was $9.7 \times 10^4$ cfu/ml, while the highest fecal and total coliform counts were 4 cfu/ml and 1,350 cfu/ml, respectively, as shown in Table 1.

Niger seed oil samples were examined, revealing a varying number of bacteria, coliforms, and/or molds. Table 2 indicates that the highest aerobic mesophilic bacterial count recorded in the examined edible oil samples was $9.2 \times 10^4$ cfu/ml, with a mean count of $3.4 \times 10^4 \pm 4 \times 10^4$ cfu/ml. Additionally, the mean total coliform count was $2.1 \times 10^2 \pm 3.5 \times 10^2$ cfu/ml (Table 2).

### Isolation and morphological characterization of isolated bacteria and fungi

Based on the macroscopic characteristics of the bacterial isolates, the A1–A5 isolates from MSA media appeared as round/cocci, yellowish colonies and were relatively small in size. The B1–B5 isolates from MAC media formed flat, smooth, colorless, and medium-sized colonies, whereas the C1–C5 isolates from MAC media exhibited mucoid, pink, and relatively small-sized colonies (Table 3).

**Table 1. Colony count from Niger seed oil samples in Gondar City, Northwest, Ethiopia (N = 12), 2021.**

| No | Name of Oil | Total aerobic bacterial count (cfu/ml) | Microbial load on MSA (cfu/ml) | Microbial load on MAC (cfu/ml) | Microbial load on PDA (cfu/ml) | Fecal coli form (cfu/ml) | Total Coli form (cfu/ml) |
|---|---|---|---|---|---|---|---|
| 1 | T1 | $1.34 \times 10^3$ | $2 \times 10^4$ | $7 \times 10^4$ | $1 \times 10^4$ | 0 | 109 |
| 2 | L | $9.2 \times 10^4$ | $1.42 \times 10^2$ | $2.21 \times 10^2$ | $2 \times 10^4$ | 2 | 1,350 |
| 3 | Y | $9.1 \times 10^4$ | 0 | $6 \times 10^4$ | 0 | 0 | 135 |
| 4 | K1 | $1.16 \times 10^3$ | $9.7 \times 10^4$ | $1.6 \times 10^3$ | $4 \times 10^5$ | 0 | 72 |
| 5 | D | $1.76 \times 10^3$ | $2.5 \times 10^3$ | $3 \times 10^4$ | $1 \times 10^4$ | 0 | 252 |
| 6 | Z | $9.2 \times 10^4$ | $9 \times 10^4$ | $1.32 \times 10^2$ | $1 \times 10^4$ | 0 | 18 |
| 7 | K2 | $7.5 \times 10^4$ | $1.1 \times 10^3$ | $4 \times 10^4$ | $3 \times 10^4$ | 0 | 63 |
| 8 | M | $8.3 \times 10^3$ | $3 \times 10^4$ | $3.3 \times 10^3$ | 0 | 4 | 21 |
| 9 | N | $3.5 \times 10^4$ | $3.2 \times 10^3$ | $1.89 \times 10^4$ | $3 \times 10^4$ | 0 | 26 |
| 10 | F | $2.1 \times 10^3$ | 0 | 0 | $4 \times 10^3$ | 2 | 173 |
| 11 | A | $1.23 \times 10^3$ | $1.5 \times 10^2$ | $3.4 \times 10^2$ | $2 \times 10^5$ | 0 | 79 |
| 12 | T2 | $4.5 \times 10^3$ | 6 | $1.5 \times 10^2$ | $1 \times 10^4$ | 3 | 312 |

**Table 2. Statistical results of different microbes of examined Niger seed oil samples in Gondar City, Northwest, Ethiopia (N = 12), 2021.**

| Type of test/organisms Minimum Maximum Mean ±Standard error | Minimum | Maximum | Mean | ±SD (standard deviation) |
|---|---|---|---|---|
| Total aerobic mesophilic bacterial plate count at 37°C/48 hours | $1.3 \times 10^3$ | $9.2 \times 10^4$ | $3.4 \times 10^4$ | $4 \times 10^4$ |
| Microbial load at 22°C/7days in PDA media | A | $4 \times 10^5$ | $6 \times 10^4$ | $1.1 \times 10^5$ |
| Microbial load in MSA media | A | $9.7 \times 10^4$ | $2.4 \times 10^4$ | $3.2 \times 10^4$ |
| Microbial load in MAC media | A | $7 \times 10^4$ | $2.1 \times 10^4$ | $2.4 \times 10^4$ |
| Total coliforms | 18 | 1,350 | $2.1 \times 10^2$ | $3.5 \times 10^2$ |
| Fecal coliforms | A | 4 | 1 | 1.5 |

A - absent.

**Table 3. Colonial, morphological characteristics of the isolated bacteria in Niger seed oil samples collected in Gondar City, Northwest, Ethiopia (N = 12), 2021.**

| Characteristics of isolated bacteria | Isolate A1-A5 from MSA | Isolate B1-B5 from MAC | Isolate C1-C5 from MAC |
|---|---|---|---|
| Colony color | Yellowish | Colorless | Pink |
| Colony shape | Round | Flat and smooth | Mucoid |
| Colony size | Small | Medium | Small |

Regarding fungal morphological characterization, the five initial fungal isolates (A1–A5) obtained from PDA media displayed large, black, powdery colonies, along with septate hyphae and elongated conidiophores. Subsequently, genus identification was performed, as outlined in Table 4.

Following the Gram reaction test, the A1–A5 and C1–C5 isolates were Gram-negative, while the B1–B5 isolates were Gram-positive. After conducting biochemical characterization, the bacteria present in the oil samples were identified at the genus level as Staphylococcus aureus, Pseudomonas aeruginosa, and Klebsiella pneumoniae (Table 5).

The microorganisms identified in the oil samples included three bacterial species, three mold species, and total and fecal coliform isolates (Table 6).

**Table 4. Cultural and morphological characteristics of the fungi iolates in Niger seed oil samples in Gondar City, Northwest, Ethiopia (N = 12), 2021.**

| Morphological Characteristics | Isolate A1-A5 on PDA | Isolate B1-B5 on PDA | Isolate C1-C5 on PDA |
|---|---|---|---|
| Cultural characteristic | Big and Yellow to green colonies | Big and Black powdery growth | Relatively big and Dark green to gray |
| Microscopic Morphology Characteristic | Conidiophores vary in length, spiny and rough septate | Septate hyphae, long conidiophores | Septate hyphae, short or long conidiophores |
| Identification underGenus | *Aspergillus flavus* | *Aspergillus niger* | *Aspergillus fumigatus* |

**Table 5. Biochemical characterization of isolated bacteria in Niger seed oil samples in Gondar City, Northwest, Ethiopia (N = 12), 2021.**

| Medias | | Isolate A1-A5 on MAC | Isolate B1-B5 on MAC | Isolate C1-C5 on MSA |
|---|---|---|---|---|
| Gram stain | Gram reaction test | -ve | +ve | -ve |
| | Cell structure | Rod-shaped | Round/ cocci | Rod-shaped |
| Biochemical test | Tsi | +ve | -ve | -ve |
| | Gas production | +ve | -ve | +ve |
| | Citrate test | +ve | +ve | +ve |
| | Urease test | -ve | +ve | +ve |
| | Motility test | +ve | -ve | -ve |
| | Starch hydrolase test | -ve | -ve | -ve |
| | Catalase test | +ve | +ve | +ve |
| | Oxidase test | +ve | -ve | -ve |
| Identification under Genus | | *Pseudomonas aeruginosa* | *Staphylococcus aureus* | *Klebsiella pneumoniae* |

(+): Positive, (-): Negative.

**Table 6. Microorganisms isolated from the Niger seed oil samples in Gondar City, Northwest, Ethiopia (N = 12), 2021.**

| Bacteria | Mold | Coliform |
|---|---|---|
| *Staphylococcus aureus* | *Aspergillus niger* | Total coliform |
| *Klebsiella pneumonia* | *Aspergillus flavus* | Fecal coliform |
| *Pseudomonas aeruginosa* | *Aspergillus fumigatus* | |

## Discussion

The quality of edible oil is ensured when manufacturing equipment, production processes, and sanitary and environmental hygiene conditions are properly maintained during processing, packaging, storage, and handling [37]. The current study assessed the microbial quality of Niger seed oils in Ethiopia. The total aerobic mesophilic bacterial plate count ranged from $1.3 \times 10^3$ to $9.2 \times 10^4$ cfu/ml. The highest mold count recorded in Niger seed oil samples was $4 \times 10^5$ cfu/ml. Total and fecal coliforms were also detected, with minimum and maximum values ranging from 4 to 1,350 cfu/ml, respectively.

In our survey, the total aerobic bacterial plate count had a minimum value of $1.3 \times 10^3$ cfu/ml and a maximum value of $9.2 \times 10^4$ cfu/ml, which is lower than the standard set by the National Agency for Food and Drug Administration and Control (NAFDAC) and the the Codex Alimentarius standard [38]. These standards specify a limit of $10^5$ cfu/ml for aerobic

mesophilic bacteria or total heterotrophic bacteria [39]. The values recorded in our study are also lower than those reported in a study conducted in Cameroon on crude palm oil, which had a maximum of $36.41 \times 10^4$ cfu/ml and a minimum of $17.14 \times 10^4$ cfu/ml [40]. However, they are slightly higher than findings from Gondar Town, where the aerobic plate count was $4.95 \times 10^3 \pm 2.76 \times 10^3$ cfu/ml [41] and a study conducted in Addis Ababa, which reported a mean count of $2.61 \times 10^4 \pm 1.47 \times 10^4$ cfu/ml [41]. Our findings are consistent with those of a Nigerian study, which reported a mean count ranging from $1.61 \times 10^4$ to $9.4 \times 10^4$ cfu/ml [39]. The observed differences in microbial counts may be attributed to variations in the types of seeds used in this study compared to previous research, which focused on oils such as olive and palm. Additionally, differences in production processes, processing methods, and storage conditions for both seeds and oils, as well as exposure to heat during marketing, may contribute to these variations. Generally, aerobic mesophilic organisms are known to cause food spoilage, including the deterioration of edible oils. Specifically, certain microorganisms produce lipase enzymes, which play a significant role in the degradation of edible oils, as suggested by [42,43].

In addition, the presence of Staphylococcus aureus, Klebsiella pneumoniae, and Pseudomonas aeruginosa was confirmed in MSA and MAC media in the oil samples, with the highest count reaching $9 \times 10^4$ cfu/ml and mean values of $3.2 \times 10^4$ cfu/ml and $2.4 \times 10^4$ cfu/ml, respectively. Bacterial contamination can accelerate oil deterioration, leading to off-flavors, rancidity, and textural changes. This reduces shelf life and overall product quality, ultimately impacting consumer satisfaction and marketability.

These bacteria are known pathogens that can cause foodborne illnesses when ingested. Staphylococcus aureus is a primary cause of food poisoning, particularly in foods that require extensive handling during preparation, which increases the risk of staphylococcal contamination. According to Levine (1938), almost all S. aureus strains produce exotoxins, and certain strains can cause food poisoning in humans upon consumption. Additionally, some data suggest that coliform bacteria may exhibit similar contamination risks [44].

Lima and colleagues [45] suggest that improper handling of products or contamination of raw materials (such as plants and seeds) may contribute to the presence of these bacteria [45]. Contamination can occur before harvest due to exposure to polluted soil, manure, irrigation water, or animal feces. These pathogens can adhere to plant leaves or penetrate plants through the endophytic root system or leaves [46,47]. Additionally, asymptomatic human carriers may introduce contamination during harvest, while post-harvest contamination can occur through contact with contaminated water or exposure to an unsanitary production environment. Beyond Staphylococcus aureus, Gram-negative bacteria such as K. pneumoniae and Pseudomonas aeruginosa are opportunistic pathogens capable of causing severe infections, including septicemia, pneumonia, urinary tract infections, and soft tissue infections. These infections pose a significant risk, particularly for individuals with weakened immune systems.

The fungal load in this study ranged from 0 to $4 \times 10^5$ cfu/ml, with a mean count of $6 \times 10^4 \pm 1.1 \times 10^5$ cfu/ml, which is higher than the acceptable fungal load limit in vegetable oils ($10^4$ cfu/ml) for aerobic mesophilic fungi, as established by NAFDAC [39] and the Codex Codex Alimentarius standard [38]. This finding is higher than that of a study conducted in Addis Ababa, which reported a mean count of $4.1 \times 10^4 \pm 4.48 \times 10^2$ cfu/ml [43], as well as studies conducted in Gondar City and by Okechalu and colleagues [39].

According to [48], molds and yeasts are spoilage microorganisms that can alter product texture and flavor, potentially leading to toxicity. Molds produce extracellular lipase enzymes, which, at high levels, can cause respiratory issues, allergic reactions, and product degradation. Moreover, molds generate mycotoxins, which pose severe health risks to both humans and animals. Similar findings in Nigeria revealed that high concentrations of yeasts and molds in edible oil result in the secretion of extracellular lipases, contributing to oil degradation [49].

As Jaya [50] further suggest, fungi play a significant role in oilseed storage, particularly under suboptimal conditions. Their study evaluated factors affecting oilseed quality during storage, emphasizing how seed mycoflora influences both the quality and quantity of oil in oilseed crops. Specifically, Actinomycetes were identified as the primary agents responsible for Niger seed infection, with occurrences reported across all types of storage containers during the storage period.

The study found the highest incidence of infection in plastic containers. Additionally, apart from Actinomycetes, five different fungal species were reported to infect Niger seeds [50].

Aspergillus niger, A. flavus, Alternaria alternata, Alternaria sp., Rhizopus nigricans, Rhizoctonia baticola, and septate sterile fungal mycelium were also reported by Siddaramiah and colleagues to be associated with Niger seed oils [16,51,52]. Additionally, Mehrotra and Aggarwal (2005) reported that A. flavus can cause tissue softening and necrosis [53]. Various studies have also examined the link between A. niger and plant diseases that cause significant financial losses [54–56]. Moreover, several studies have documented that Aspergillus species are capable of producing mycotoxins, which can lead to a range of health issues affecting multiple bodily systems, including the liver, kidneys, nervous system, skin, respiratory system, and digestive organs. Additionally, these fungi are known to produce aflatoxins, ochratoxin A, and fumonisin B2 when present in stored oil and other food commodities.

In the current study, coliforms were detected in the majority of oil samples. Total coliforms were found to be within a range of 18–1,350 cfu/ml, with a mean value of $2.1 \times 10^2 \pm 3.5 \times 10^2$ cfu/ml. This result exceeds the standard for edible vegetable oils, which is $10^2$ cfu/ml for coliforms [39]. The presence of coliform bacteria in edible oils is a significant indicator of microbial contamination, suggesting potential health risks and compromised product quality. Their detection indicates possible contamination from environmental sources or during processing. However, this value is lower than that reported in a study conducted in Addis Ababa [43], which found a mean value of $6.6 \times 10^4 \pm 2.62 \times 10^2$ cfu/ml [43]. Fecal coliforms were present in four oil samples, which is lower than findings from other studies, where values of $1.6 \times 10^4$ and $3.28 \times 10^3$ cfu/ml were reported [43]. Since coliforms originate from sources similar to those of pathogenic organisms, they respond similarly to environmental conditions and treatments, often serving as indicators of pathogen presence. The detection of fecal coliforms in edible oils often points to unhygienic handling, processing, or storage practices. However, research suggests that food poisoning is not primarily attributed to coliforms, as humans possess a certain level of resistance to the toxins produced by coliform bacteria. Instead, coliform concentration is considered an indicator of overall hygienic conditions during oil production and handling, as well as the effectiveness of contamination control measures [39,57,58].

Nevertheless, high levels of coliforms can negatively impact the sensory attributes and shelf life of the oil, leading to deterioration in quality and decreased consumer satisfaction. Additionally, the presence of coliform bacteria in Niger seed oil can harm the product's reputation and the brand associated with it. Consumers may lose trust in the safety and quality of the oil, resulting in decreased demand and potential economic losses for producers and manufacturers.

### Strength and limitation

This study provides valuable baseline data on the microbial quality of Niger seed oil in Ethiopia, addressing a critical knowledge gap. It employs standardized and rigorous microbiological methods for isolation, enumeration, and identification of key bacterial and fungal contaminants, ensuring reliable results. The findings are contextualized through comparisons with national and international standards, underscoring public health relevance. By identifying potentially harmful microorganisms, the study offers important insights for food safety stakeholders and lays a foundation for improving hygienic practices and quality control in edible oil production and distribution.

A limitation of this study is its cross-sectional design, which reflects microbial contamination at a single point in time and does not capture possible seasonal or temporal variations. The analysis primarily focused on culturable bacterial and fungal contaminants, without the use of molecular techniques or assessment of other microbial pathogens and mycotoxins. Future research incorporating molecular methods could offer more precise identification and enhanced safety evaluation. Additionally, longitudinal studies would be valuable to observe changes in contamination over time.

### Conclusions

Foodborne diseases and instances of food poisoning stemming from microbial contamination of edible seed oils are a significant global public health and food safety concern. Effective regulation of these issues becomes more manageable

when there is a clear understanding of the quality of edible seed oils and regular assessments thereof. The microorganisms isolated in this study encompass indicator organisms, fungi, and potentially harmful bacteria. Any detection of these microorganisms at any point along the processing, packaging, and retail stages indicates that the oil may not meet cleanliness standards. Ingesting this contaminated edible oil can pose health risks, exacerbated by the presence of food-spoiling organisms, particularly molds, which accelerate the oil's degradation. This underscores the importance and requirement of implementing stringent hygiene practices, quality control measures, and adherence to food safety regulations throughout the production and distribution process through coordinated efforts to enhance the safety of edible oil in Ethiopia. Additionally, local manufacturers should be subjected to stringent quality and safety control measures, while investors should be encouraged through incentives to develop and use innovative and updated technologies for edible oil production.

## Implication of the study

This study highlights critical public health concerns related to the microbial contamination of Niger seed oil in Gondar City, emphasizing the need for stringent hygiene practices and regulatory oversight. The detection of pathogenic bacteria and molds suggests potential risks to consumer health and underscores the importance of routine quality control and food safety monitoring. These findings provide valuable evidence to inform policymakers, producers, and health authorities in developing targeted interventions to improve edible oil safety, thereby protecting public health and enhancing market confidence in locally produced oils.

## Acknowledgments

The authors are grateful for the University of Gondar, Gondar Trade and Industry office, study participants, and facilitators.

## Author contributions

**Conceptualization:** Lamrot Yohannes.

**Data curation:** Lamrot Yohannes, Mastewal Endalew.

**Formal analysis:** Lamrot Yohannes, Fasika Weldegebrel.

**Funding acquisition:** Lamrot Yohannes.

**Investigation:** Lamrot Yohannes.

**Methodology:** Lamrot Yohannes, Mastewal Endalew.

**Project administration:** Lamrot Yohannes, Tsegaye Adane Birhan.

**Resources:** Lamrot Yohannes, Fasika Weldegebrel.

**Software:** Lamrot Yohannes.

**Supervision:** Lamrot Yohannes, Jember Azanaw.

**Validation:** Lamrot Yohannes, Jember Azanaw.

**Visualization:** Lamrot Yohannes, Tsegaye Adane Birhan.

**Writing – original draft:** Lamrot Yohannes.

**Writing – review & editing:** Lamrot Yohannes, Tsegaye Adane Birhan.

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
