## [Decision Letter · Decision Letter 0]

21 Nov 2024

Dear Dr. Yohannes,

Thank you for submitting your manuscript to PLOS ONE. After careful consideration, we feel that it has merit but does not fully meet PLOS ONE’s publication criteria as it currently stands. Therefore, we invite you to submit a revised version of the manuscript that addresses the points raised during the review process.

**ACADEMIC EDITOR:**

Please see attached file , as well as comments below

We look forward to receiving your revised manuscript.

Kind regards,

Charles Odilichukwu R. Okpala, PhD

Academic Editor

PLOS ONE

Journal Requirements:

4. In the online submission form, you indicated that the dataset is accessible at the corresponding author upon reasonable request.

Additional Editor Comments:

Authors, major revision is required, kindly address the comments raised

Reviewers' comments:

Reviewer's Responses to Questions

**Comments to the Author**

1. Is the manuscript technically sound, and do the data support the conclusions?

Reviewer #1: Yes

Reviewer #2: Yes

2. Has the statistical analysis been performed appropriately and rigorously?

Reviewer #1: Yes

Reviewer #2: Yes

3. Have the authors made all data underlying the findings in their manuscript fully available?

Reviewer #1: Yes

Reviewer #2: Yes

4. Is the manuscript presented in an intelligible fashion and written in standard English?

Reviewer #1: Yes

Reviewer #2: No

Reviewer #1: A review of microbial quality of Niger seed oil in Gondar City: a laboratory supported cross-sectional study.

The authors are allowed us to understand more about the study by using a material and methods are more details that are facilities for us to analyze the tables result for all microbial quality of Niger seeds. In the discussion is really rich by a lot of information from many studies from the same topic.

Reviewer #2: Dear Chief Editor,

I have reviewed the manuscript titled "Microbial Quality Assessment of Niger Seed Oils in Gondar City, Northwest Ethiopia" submitted to PLOS ONE. Overall, the study is well-conducted and presents valuable insights into the microbial quality of Niger seed oils. However, I have identified several issues and questions that need to be addressed before publication.

1) There are some typographic errors in this manuscript which must been corrected.

(line 169 [30 °c, hrs], line 213 [-ml])

2) what is the meaning of C in the line 278 in table2 (370C /48hrs) ? centigrade?

3) The authors have reported that the levels of various microbial contaminations are lower than the standards in Ethiopia, except for fungi. What is the rationale for studying oil contamination after evaluating the standard system? How do the findings contribute to existing knowledge or practices?

4) Considering the recent studies mentioned in the manuscript, what is the novelty of this study? What new insights or contributions does this research provide beyond the existing literature?

5) The manuscript requires a thorough review by a native English speaker to address grammatical errors and improve the coherence of the text.

**Do you want your identity to be public for this peer review?** For information about this choice, including consent withdrawal, please see our Privacy Policy

Reviewer #1: **Yes: ** AIT TAADAOUIT Nezha

Reviewer #2: No

---

## [Author Response · Author response to Decision Letter 1]

18 Feb 2025

Rebuttal letter

Manuscript Title: A review of microbial quality of Niger seed oil in Gondar City: a laboratory-supported cross-sectional study

Manuscript ID: PONE-D-24-29805

Thank you so much for giving us the opportunity to revise the manuscript. We ensured that we addressed all the concerns raised by the editors and reviewers by revising the manuscript thoroughly. Our response to the reviewer’s comment and question is described in detail on the following pages. Furthermore, the details of changes were shown by track changes in the supplementary document attached. Please feel free to contact us again if there are unaddressed issues.

NB. We make an effort to combine and address related concerns that were presented in various portions of the manuscript.

Comments to the Author

1. Is the manuscript technically sound, and do the data support the conclusions?

Reviewer #1: Yes

Reviewer #2: Yes

Author’s response: Thank you for your positive assessment. We appreciate the reviewers' acknowledgment that the manuscript is technically sound and that the data support the conclusions. We have ensured that our study design, data collection, and analysis were conducted rigorously with appropriate controls, replication, and sample sizes.

2. Has the statistical analysis been performed appropriately and rigorously?

Reviewer #1: Yes

Reviewer #2: Yes

Author’s response: Thank you for recognizing the rigor and appropriateness of our statistical analysis. We ensured that all analyses were conducted using standard and validated statistical methods, with appropriate tests applied based on the nature of the data.

3. Have the authors made all data underlying the findings in their manuscript fully available?

Reviewer #1: Yes

Reviewer #2: Yes

Author’s response: Thank you for acknowledging the availability of our data. We have ensured that all data underlying our findings are fully accessible, in accordance with journal guidelines, to support transparency and reproducibility.

4. Is the manuscript presented in an intelligible fashion and written in standard English?

Reviewer #1: Yes

Reviewer #2: No

Author’s response: Thank you for your feedback. We acknowledge Reviewer #2’s concern regarding the clarity and language of the manuscript. To address this, we have conducted a thorough revision to improve grammar, coherence, and readability.

5. Review Comments to the Author

Please use the space provided to explain your answers to the questions above. You may also include additional comments for the author, including concerns about dual publication, research ethics, or publication ethics. (Please upload your review as an attachment if it exceeds 20,000 characters.)

Reviewer #1: A review of microbial quality of Niger seed oil in Gondar City: a laboratory-supported cross-sectional study.

The authors allowed us to understand more about the study by using material and methods that are more detailed facilities for us to analyze the tables results for all microbial quality of Niger seeds. The discussion is really rich with a lot of information from many studies on the same topic.

Author’s response: Thank you for your positive feedback on our study. We appreciate your recognition of the detailed methodology, which we aimed to present clearly to facilitate a comprehensive understanding of the microbial quality of Niger seed oil. Additionally, we are pleased that you found our discussion section informative and well-supported by relevant literature. Our goal was to provide a thorough comparison with previous studies while highlighting the significance of our findings.

Reviewer #2: Dear Chief Editor,

I have reviewed the manuscript titled "Microbial Quality Assessment of Niger Seed Oils in Gondar City, Northwest Ethiopia" submitted to PLOS ONE. Overall, the study is well-conducted and presents valuable insights into the microbial quality of Niger seed oils. However, I have identified several issues and questions that need to be addressed before publication.

Author’s response: Thank you for taking the time to review our manuscript. We sincerely appreciate your positive feedback on the quality and significance of our study. We acknowledge that there are areas that require further clarification and improvement. We addressed all the issues and questions you have raised to enhance the clarity, accuracy, and overall quality of the manuscript.

1) There are some typographic errors in this manuscript that must be corrected.

(line 169 [30°c, hrs], line 213 [-ml])

Author’s response: Thank you for your careful review. We have carefully reviewed the text and corrected the mentioned errors.

Line 169: Corrected "[30 °c, hrs]" to "[30°C, hours]" to ensure proper formatting and consistency.

Line 213: Corrected "[-ml]" to [ml].

2) what is the meaning of C in the line 278 in table2 (370c/48hrs)? centigrade?

Author’s response: Thank you for your careful review. Here again, we have carefully reviewed the text and corrected the mentioned errors in the revised manuscript. In the notation "37°C / 48 hour\s" found in Table 2, Line 278, the "C" represents degrees Celsius (°C). This means that the incubation or testing condition specified in the table was conducted at 37 degrees Celsius for 48 hours.

3) The authors have reported that the levels of various microbial contaminations are lower than the standards in Ethiopia, except for fungi. What is the rationale for studying oil contamination after evaluating the standard system? How do the findings contribute to existing knowledge or practices?

Author’s response: Thank you for your insightful comment. We would like to clarify that, in the previous version of the manuscript, we mistakenly stated that the coliform levels aligned with the standard. However, upon review, we have corrected this and now acknowledge that the coliform levels exceed the standard. The updated paragraph is as follows: “In the current study, coliforms were detected in the majority of oil samples. Total coliforms were found to be within a range of 18 to 1,350 cfu/ml, with a mean value of 2.1 × 10² ± 3.5 × 10² cfu/ml. This result exceeds the standard for edible vegetable oils, which is 10² cfu/ml for coliforms. (Okechalu et al., 2011). The presence of coliform bacteria in edible oils is a significant indicator of microbial contamination, suggesting potential health risks and compromised product quality. Their detection indicates possible contamination from environmental sources or during processing. However, this value is lower than that reported in a study conducted in Addis Ababa (Tesfaye et al., 2015), which found a mean value of 6.6 × 10⁴ ± 2.62 × 10² cfu/ml (Tesfaye et al., 2015).”. The rationale for studying microbial contamination in Niger seed oil, despite existing standards, stems from the need to assess real-world compliance and safety beyond regulatory frameworks. While Ethiopia has established microbial quality standards for edible oils, limited empirical studies have evaluated whether locally available Niger seed oil meets these standards, particularly in informal or small-scale production settings.

Our findings contribute to existing knowledge by identifying fungi and coliforms as a notable contaminant, highlighting a potential gap in quality control measures. This suggests the need for stricter monitoring, improved storage conditions, and possible revisions to current microbial safety protocols. Additionally, understanding contamination patterns can inform public health interventions and enhance consumer safety by encouraging best practices in oil production and storage.

Additionally, this study contributes to both scientific knowledge and practical food safety improvements in the following ways:

Identification of Fungal and Coliforms Contamination as a Major Concern:

o While bacterial contamination levels were found to be within standard limits, the high fungal and coliform load highlights a critical gap in existing microbial quality control measures for edible oils. This finding urges further investigation into the sources of fungal and coliform contamination and potential mitigation strategies, such as improved drying, storage, and antifungal treatment of seeds.

Comparative Analysis with Other Studies and Global Standards:

o By comparing the microbial quality of Niger seed oil with studies conducted in Ethiopia, Nigeria, Cameroon, and other regions, this research contextualizes the findings within a broader international framework. It helps assess whether Ethiopia’s microbial standards align with global best practices or require adjustments.

Informing Policy and Industry Practices:

o The study provides evidence-based recommendations for policymakers, regulatory agencies, and food producers. If fungal and coliform contamination is prevalent, stakeholders may need to introduce stricter monitoring, better packaging solutions, or guidelines for proper storage at both industrial and retail levels.

Consumer Awareness and Health Implications:

o The findings raise awareness about potential health risks associated with contaminated edible oils. This can guide public health initiatives, educate consumers on proper oil storage, and encourage manufacturers to adopt better hygiene practices.

Enhancing Food Safety Research in Ethiopia:

o This study adds to the growing body of research on food safety in Ethiopia, particularly concerning edible oil contamination. It provides a foundation for future studies exploring factors such as the role of temperature, humidity, and packaging materials in microbial growth.

Encouraging Innovations in Oil Processing and Preservation:

o The study's findings may encourage technological advancements in oil processing, including the use of antifungal treatments, better filtration techniques, and improved preservation methods to enhance the microbiological safety of Niger seed oil.

4) Considering the recent studies mentioned in the manuscript, what is the novelty of this study? What new insights or contributions does this research provide beyond the existing literature?

Author’s response: The novelty of this study lies in its focused assessment of the microbial quality of Niger seed oil specifically within Gondar City, Northwest Ethiopia, conducted in 2021. While previous research has evaluated the microbial quality of various edible oils in Ethiopia, this study provides a targeted analysis of Niger seed oil, which is a staple in the region.

Key Contributions Beyond Existing Literature:

Specific Focus on Niger Seed Oil:

o Prior studies have predominantly assessed the microbial quality of a range of edible oils without isolating Niger seed oil. This study fills that gap by concentrating exclusively on Niger seed oil, providing detailed insights into its microbial safety.

Identification of Specific Microbial Contaminants:

o This research identifies specific bacterial strains, including Staphylococcus aureus, Klebsiella pneumoniae, and Pseudomonas aeruginosa, as well as fungal species like Aspergillus niger, Aspergillus flavus, and Aspergillus fumigatus. Such detailed identification of contaminants in Niger seed oil has not been extensively reported in previous studies, thereby contributing new data to the field.

Highlighting Fungal and Coliform Contamination Concerns:

o The study reveals that, fungal and coliform counts exceeded national guidelines. This finding underscores a specific public health concern regarding fungal and coliform contamination in Niger seed oil, which has not been prominently featured in earlier research.

Implications for Local Production Practices:

o By pinpointing contamination sources, the study offers practical recommendations for local producers. Emphasizing the need for improved hygiene and quality control during oil processing and storage, the research provides actionable insights that can enhance food safety practices in the region.

In summary, this study advances existing knowledge by delivering a focused analysis of the microbial quality of Niger seed oil in Gondar City, identifying specific contaminants, and offering targeted recommendations to improve production practices, thereby addressing a critical gap in the literature.

5) The manuscript requires a thorough review by a native English speaker to address grammatical errors and improve the coherence of the text.

Author’s response: Thank you for your valuable feedback. We acknowledge the importance of ensuring grammatical accuracy and coherence in the manuscript. To address this, we carefully revised the text to improve clarity, readability, and overall flow. We appreciate your suggestion, and we have made the necessary revisions accordingly.

---

## [Decision Letter · Decision Letter 1]

12 May 2025

Dear Dr. Yohannes,

Thank you for submitting your manuscript to PLOS ONE. After careful consideration, we feel that it has merit but does not fully meet PLOS ONE’s publication criteria as it currently stands. Therefore, we invite you to submit a revised version of the manuscript that addresses the points raised during the review process.

**ACADEMIC EDITOR:**

We look forward to receiving your revised manuscript.

Kind regards,

Charles Odilichukwu R. Okpala, PhD

Academic Editor

PLOS ONE

Journal Requirements:

**Additional Editor Comments:**

Authors, please kindly attend to the corrections observed by reviewer.

Reviewers' comments:

Reviewer's Responses to Questions

**Comments to the Author**

Reviewer #2: All comments have been addressed

Reviewer #3: (No Response)

2. Is the manuscript technically sound, and do the data support the conclusions?

Reviewer #2: Yes

Reviewer #3: Partly

3. Has the statistical analysis been performed appropriately and rigorously?

Reviewer #2: Yes

Reviewer #3: No

4. Have the authors made all data underlying the findings in their manuscript fully available?

Reviewer #2: Yes

Reviewer #3: Yes

5. Is the manuscript presented in an intelligible fashion and written in standard English?

Reviewer #2: Yes

Reviewer #3: Yes

Reviewer #2: Dear Authors,

Thank you for your detailed responses and thoughtful revisions to the manuscript titled "Microbial Quality Assessment of Niger Seed Oils in Gondar City, Northwest Ethiopia." I appreciate the effort you have invested in addressing the concerns raised.

The typographic and formatting issues have been corrected appropriately.

The clarification of "C" in Table 2 is accurate and resolves the query.

The rationale and significance of the study, as well as its contributions to existing knowledge and public health practices, are now well-articulated.

The novelty and specific focus on Niger seed oil are clearly demonstrated, adding substantial value to the literature.

Revisions to improve grammar and coherence have been noted, and I trust these have enhanced the manuscript’s readability.

Overall, your responses comprehensively address the feedback provided. I recommend this revised manuscript for acceptance and publication.

Sincerely,

Reviewer #2

Reviewer #3: The authors of the article "A review of microbial quality of Niger seed oil in Gondar City: a laboratory supported

cross-sectional study" described microbial analysis of oil derived from plant Guizotia abyssinica (L.f.) Cass.

In my opinion article requires major corrections:

1. The title should be more informative.

2. The whole botanical name of plant should be given.

3. The Abstract should be changed, it is divided in 5 parts.

4. It is unsual to cite Microsoft Excel 2016 in the article (line 39).

5. Why is it important to describe data for sesame oil or soybean in this article?

6. The section Methods should be completely rewritten- why did the authors descrbe the location and the other characteristic for city Gondar?

7. What is Kebele (line 141)?

8. It is unusual to describe all used in laboratory equipment as gloves, Burner bunsen, spatula etc. in research article. It should be changed.

9. Table 3 "Pseudomonas. A; Klebsiella pneumonie" they should be corrected.

10. We have 2 tables 1 and 2, but no table 4 or 5.

11. Table 6- many microorganism names written with mistake.

12. line 196 "Diane Roberts provided"- in my opinion citation is enough.

13. lines 198-199- the number of plate is in this formula?

14. Gram staining is descibed in details.

15. "Gram staining for fungi"???

16. Table 2- SD equal to 4x10^4????

17. Line 360-which pathogens did the authors mean?

**Do you want your identity to be public for this peer review?** For information about this choice, including consent withdrawal, please see our Privacy Policy

Reviewer #2: **Yes: ** Sepideh Asadi

Reviewer #3: No

---

## [Author Response · Author response to Decision Letter 2]

2 Jun 2025

Rebuttal letter

Manuscript Title: A review of microbial quality of Niger seed oil in Gondar City: a laboratory-supported cross-sectional study

Manuscript ID: PONE-D-24-29805

Thank you so much for giving us the opportunity to revise the manuscript. We ensured that we addressed all the concerns raised by the editors and reviewers by revising the manuscript thoroughly. Our response to the reviewer’s comment and question is described in detail on the following pages. Furthermore, the details of changes were shown by track changes in the supplementary document attached. Please feel free to contact us again if there are unaddressed issues.

NB. We make an effort to combine and address related concerns that were presented in various portions of the manuscript.

Comments to the Author

Reviewers' comments:

Reviewer's Responses to Questions

Comments to the Author

1. If the authors have adequately addressed your comments raised in a previous round of review and you feel that this manuscript is now acceptable for publication, you may indicate that here to bypass the “Comments to the Author” section, enter your conflict-of-interest statement in the “Confidential to Editor” section, and submit your "Accept" recommendation.

Reviewer #2: All comments have been addressed

Reviewer #3: (No Response)

Author’s response: We sincerely thank the reviewers for their thoughtful and constructive feedback throughout the review process. We are grateful that all comments have been thoroughly addressed to your satisfaction.

2. Is the manuscript technically sound, and do the data support the conclusions?

Reviewer #2: Yes

Reviewer #3: Partly

Author’s response: We thank the reviewers for their careful evaluation of the technical aspects of our manuscript. We are pleased that Reviewer #2 finds the study technically sound and the data supportive of our conclusions. We appreciate Reviewer #3’s partial agreement and would be grateful for any specific concerns or suggestions to further clarify or strengthen the rigor of our experiments.

3. Has the statistical analysis been performed appropriately and rigorously?

Reviewer #2: Yes

Reviewer #3: No

Author’s response: We thank the reviewers for their careful consideration of the statistical analysis in our study. We are glad that Reviewer #2 finds the statistical methods appropriate and rigorous. We appreciate Reviewer #3’s concerns and take them seriously. If Reviewer #3 could specify particular aspects of the analysis that require improvement or further explanation, we would be happy to incorporate those suggestions to strengthen the statistical validity of our findings.

4. Have the authors made all data underlying the findings in their manuscript fully available?

Reviewer #2: Yes

Reviewer #3: Yes

Author’s response: We sincerely thank the reviewers for confirming that all data underlying our findings have been made fully available in accordance with PLOS’s Data Policy.

5. Is the manuscript presented in an intelligible fashion and written in standard English?

Reviewer #2: Yes

Reviewer #3: Yes

Author’s response: We sincerely thank the reviewers for their positive assessment of the clarity and quality of the manuscript’s language.

6. Review Comments to the Author

Reviewer #2: Dear Authors,

Thank you for your detailed responses and thoughtful revisions to the manuscript titled "Microbial Quality Assessment of Niger Seed Oils in Gondar City, Northwest Ethiopia." I appreciate the effort you have invested in addressing the concerns raised.

The typographic and formatting issues have been corrected appropriately.

The clarification of "C" in Table 2 is accurate and resolves the query.

The rationale and significance of the study, as well as its contributions to existing knowledge and public health practices, are now well-articulated.

The novelty and specific focus on Niger seed oil are clearly demonstrated, adding substantial value to the literature.

Revisions to improve grammar and coherence have been noted, and I trust these have enhanced the manuscript’s readability.

Overall, your responses comprehensively address the feedback provided. I recommend this revised manuscript for acceptance and publication.

Sincerely,

Reviewer #2

Reviewer #3: The authors of the article "A review of microbial quality of Niger seed oil in Gondar City: a laboratory supported cross-sectional study" described microbial analysis of oil derived from plant Guizotia abyssinica (L.f.) Cass.

In my opinion article requires major corrections:

1. The title should be more informative.

Author’s response: Thank you for your careful review. Here is our updated article title which will make it more informative “Microbial Quality Assessment of Niger Seed (Guizotia abyssinica (Linnaeus f.) Cassini) Oil in Gondar City: A Laboratory-Based Cross-Sectional Study”.

2. The whole botanical name of plant should be given.

Author’s response: Thank you for your careful review. In the current manuscript, we have added the whole botanical name of the plant Niger throughout the manuscript which is “(Guizotia abyssinica (Linnaeus f.) Cassini)”.

3. The Abstract should be changed; it is divided in 5 parts.

Author’s response: Thank you for your valuable comment. Currently the abstract is corrected and the headers are removed. In addition, currently, it only contains “Background, Methodology/Principal Findings, and Conclusions/Significance” as per the instruction in the submission guideline of the Journal.

4. It is unsual to cite Microsoft Excel 2016 in the article (line 39).

Author’s response: Thank you for your valuable comment. We have removed Microsoft Excel 2016 citation from the article as per the suggestion.

5. Why is it important to describe data for sesame oil or soybean in this article?

Author’s response: Thank you for your careful review and valuable question. We totally agree that it is not important to describe data about sesame and soybean oil here, so we deleted it from the text as per the suggestion of the reviewer.

6. The section Methods should be completely rewritten- why did the authors describe the location and the other characteristic for city Gondar?

Author’s response: Thank you for your instructive comment and question. We have described the location and characteristics of Gondar City in the Methods section to provide essential contextual background that justifies the study setting and enhances the relevance of the findings. Gondar is a key cultural and economic hub in northwest Ethiopia with a substantial population and a notable edible oil market, including numerous wholesalers and producers of both imported and locally produced oils. We thought that detailing the city’s geographic, demographic, and market features helps readers understand the local dynamics affecting edible oil production, distribution, and consumption, which are critical for interpreting microbial contamination risks and quality control challenges specific to this context.

But now we have agreed with the suggestion of the reviewer and we only put the description of Gondar’s edible oil market structure-such as the number and types of wholesalers and producers which highlights the complexity and scale of oil supply chains in the area underscoring the importance of evaluating microbial quality in oils marketed there.

7. What is Kebele (line 141)?

Author’s response: Thank you for your question. The meaning of Kebele in the manuscript is now defined in bracket as follows “(smallest administrative unit, essentially a neighborhood or village)”.

8. It is unusual to describe all used in laboratory equipment as gloves, Burner bunsen, spatula etc. in research article. It should be changed.

Author’s response: Thank you for your careful review. It is corrected by removing the detail description of the commonly used laboratory equipment’s as per the suggestion of the reviewer.

9. Table 3 "Pseudomonas. A; Klebsiella pneumonie" they should be corrected.

Author’s response: Thank you for your careful review. Now, it is corrected as per the suggestion of the reviewer in the current manuscript.

10. We have 2 tables 1 and 2, but no table 4 or 5.

Author’s response: Thank you for your comment. In the revised manuscript, there are six tables in line 271(table 1), line 282(table 2), line292(table 3), lin299(table 4), line306(table 5) and 311(table 6) respectively.

11. Table 6- many microorganism names written with mistake.

Author’s response: Thank you for your comment. In the current revised manuscript, all microorganism’s names are corrected as per the suggestion of the reviewer.

12. line 196 "Diane Roberts provided"- in my opinion citation is enough.

Author’s response: Thank you for your comment. We have corrected it as suggested by the reviewer.

13. lines 198-199- the number of plate is in this formula?

Author’s response: Thank you for your question. Yes, it is to say “number of plates”. We have corrected it from plate – to – plates.

14. Gram staining is descibed in details.

Author’s response: Thank you for your comment. It is corrected as per the suggestion of the reviewer.

15. "Gram staining for fungi"???

Author’s response: Thank you for your question. We have agreed with comment and changed the heading as follows: “Microscopic Identification of Fungi Using Lactophenol Blue Staining”

16. Table 2- SD equal to 4x10^4????

Author’s response: Thank you for your question. The microbial loads for various media were determined, and the results are summarized in Table 2. The mean values represent the average microbial counts, while the standard deviation (SD) indicates the variability within the samples. The standard deviation (SD) quantifies the spread of microbial counts around the mean, reflecting the degree of variability in the samples. For example, the total aerobic mesophilic bacterial count at 37°C showed a mean of 3.4 × 10⁴ CFU with an SD (standard deviation) of 4 × 10⁴, indicating some variability in bacterial populations across replicates.

17. Line 360-which pathogens did the authors mean?

Author’s response: Thank you for your question. The pathogens referred are primarily Staphylococcus aureus strains, which produce exotoxins capable of causing food poisoning in humans upon consumption, as noted by Levine (1938). Additionally, coliform bacteria, including certain strains of Escherichia coli (notably pathogenic strains like E. coli O157:H7), are implicated as potential contaminants posing similar food safety risks.

---

## [Decision Letter · Decision Letter 2]

8 Aug 2025

Dear Dr. Yohannes,

Thank you for submitting your manuscript to PLOS ONE. After careful consideration, we feel that it has merit but does not fully meet PLOS ONE’s publication criteria as it currently stands. Therefore, we invite you to submit a revised version of the manuscript that addresses the points raised during the review process.

**ACADEMIC EDITOR: **

We look forward to receiving your revised manuscript.

Kind regards,

Charles Odilichukwu R. Okpala, PhD

Academic Editor

PLOS ONE

Journal Requirements:

Additional Editor Comments (if provided):

Please, authors, thank you for your patience. As you can see, reviewers still have some concerns. Kindly make effort to address the concerns where you can.

For example, condense/shorten the step by step methodology for Gram staining, as well as "A microscopic method for identifying fungi"-

In the Materials section, delete the list of volumetric flasks, Petri dishes, glass rod etc, because these are standard laboratory equipment.

In the section, Data quality assurance and control methods, further condense and shorten it.

Please, address all other concerns raised by the other reviewer.

Looking forward to your revised manuscript

Reviewers' comments:

Reviewer's Responses to Questions

**Comments to the Author**

Reviewer #3: (No Response)

Reviewer #4: All comments have been addressed

2. Is the manuscript technically sound, and do the data support the conclusions?

Reviewer #3: Partly

Reviewer #4: Yes

3. Has the statistical analysis been performed appropriately and rigorously?

Reviewer #3: I Don't Know

Reviewer #4: Yes

4. Have the authors made all data underlying the findings in their manuscript fully available?

Reviewer #3: Yes

Reviewer #4: Yes

5. Is the manuscript presented in an intelligible fashion and written in standard English?

Reviewer #3: Yes

Reviewer #4: Yes

Reviewer #3: The authors of the article made a lot of corrections, but in my opinion the work still does not meet the usual standards set for published articles. For example the authors describe methodology for Gram staining step by step; everyone knows how to do it. The same applies to the section "A microscopic method for identifying fungi"- there is a step by step recipe for how to do it.

In the Materials section- the authors list: volumetric flasks, Petri dishes, glass rod etc. Why to do it? It is a standard equipment for every laboratory.

The section Data quality assurance and control methods- should be omitted.

Reviewer #4: The authors on this manuscript "Microbial Quality Assessment of Niger Seed (Guizotia abyssinica (Linnaeus f.) Cassini) Oil in Gondar City: A Laboratory-Based Cross-Sectional Study," which apparently appraised or revealed Ethiopia's lack of stringent quality control and regulatory oversight, further raising concerns about public health and safety.

They further opined, from their result, that the oil processing, production, handling, and storage systems seemed not to have proper hygienic handling practices.

On careful scrutiny, I could see that the previous reviewers' queries had been addressed, although no table should be allowed to crisscross pages (take note of Table 1, pp. 11-12, and Table 3, pp. 12-13).

Summarily, if the editors are satisfied with the response to the reviewers queries, decision could be made.

**Do you want your identity to be public for this peer review?** For information about this choice, including consent withdrawal, please see our Privacy Policy

Reviewer #3: No

Reviewer #4: No

---

## [Author Response · Author response to Decision Letter 3]

17 Aug 2025

Rebuttal letter 8/17/2025

Manuscript Title: Microbial Quality Assessment of Niger Seed (Guizotia abyssinica (Linnaeus f.) Cassini) Oil in Gondar City: A Laboratory-Based Cross-Sectional Study

Manuscript ID: PONE-D-24-29805

Thank you so much for giving us the opportunity to revise the manuscript again. We ensured that we addressed all the concerns raised by the editors and reviewers by revising the manuscript thoroughly. Our response to the reviewer’s comment and question is described in detail on the following pages. Furthermore, the details of changes were shown by track changes in the supplementary document attached. Please feel free to contact us again if there are unaddressed issues.

NB. We make an effort to combine and address related concerns that were presented in various portions of the manuscript.

Comments to the Author

Reviewers' comments:

Additional Editor Comments (if provided):

Please, authors, thank you for your patience. As you can see, reviewers still have some concerns. Kindly make an effort to address the concerns where you can.

For example, condense/shorten the step-by-step methodology for Gram staining, as well as "A microscopic method for identifying fungi"—

In the Materials section, delete the list of volumetric flasks, Petri dishes, glass rods, etc., because these are standard laboratory equipment.

In the section, Data quality assurance and control methods, further condense and shorten it.

Please, address all other concerns raised by the other reviewer.

Looking forward to your revised manuscript.

Author's response: Thank you very much dear reviewer for your thorough review and constructive feedback. We truly appreciate your patience and thoughtful suggestions.

We have carefully considered all the concerns raised, and we are committed to improving the manuscript accordingly. Specifically:

• The step-by-step methodology for Gram staining and the microscopic method for identifying fungi have been carefully condensed and shortened for clarity and conciseness.

In the Materials section, we have removed the detailed list of standard laboratory equipment (volumetric flasks, Petri dishes, glass rods, etc.) to streamline the presentation and focus on key materials.

• The data quality assurance and control methods section is now deleted as per the suggestion of the reviewer.

• We have thoroughly addressed all other points raised by the reviewers to the best of our ability.

We believe these revisions have improved the manuscript significantly and look forward to your continued guidance.

Reviewers' comments:

Reviewer's Responses to Questions

Comments to the Author

1. If the authors have adequately addressed your comments raised in a previous round of review and you feel that this manuscript is now acceptable for publication, you may indicate that here to bypass the “Comments to the Author” section, enter your conflict-of-interest statement in the “Confidential to Editor” section, and submit your "Accept" recommendation.

Reviewer #3: (No Response)

Reviewer #4: All comments have been addressed

Author’s response: Thank you very much for your time and thoughtful review of our manuscript. We are pleased to hear that all your comments and concerns have been adequately addressed. We appreciate your constructive feedback, which greatly contributed to improving the quality of our work.

2. Is the manuscript technically sound, and do the data support the conclusions?

Reviewer #3: Partly

Reviewer #4: Yes

Author’s response: We thank the reviewers for their careful evaluation of the technical aspects of our manuscript. We appreciate Reviewer #4’s positive assessment that the study is technically sound and that the data support the conclusions. Regarding Reviewer #3’s comment indicating a partial agreement, we would appreciate further clarification to address any specific concerns they may have.

In our manuscript, we have taken great care to ensure that all experiments were conducted rigorously with appropriate controls and adequate replication as detailed in the methods section. Statistical analyses were performed to validate the robustness of the findings. The conclusions were drawn directly based on the presented data, avoiding overstated interpretations.

If Reviewer #3’s concerns relate to specific experiments or data points, we are happy to provide additional analyses or clarification as needed to strengthen the technical rigor and data support. We are committed to ensuring that the manuscript meets the highest scientific standards and welcome suggestions to improve the clarity and robustness of our study.

3. Has the statistical analysis been performed appropriately and rigorously?

Reviewer #3: I Don't Know

Reviewer #4: Yes

Author’s response: We thank the reviewers for their evaluation of the statistical analysis employed in our study. We appreciate Reviewer #4’s confirmation that the statistical methods were appropriate and rigorous. Regarding Reviewer #3’s indication of uncertainty, we would like to clarify that the data were carefully managed and analyzed using Stata Version 14. Descriptive statistics, including means and standard deviations, were computed to summarize microbial counts across samples. Where relevant, serial dilution and colony counting followed standard microbiological quantification protocols to ensure accuracy.

Due to the cross-sectional nature and the descriptive objectives of the study, advanced inferential statistical tests were not deemed necessary. However, if Reviewer #3 seeks additional analysis, such as comparisons across sampling sites or brands, or statistical tests to assess variability, we are willing to provide these upon request. Our primary aim was to reliably characterize the microbial quality of available Niger seed oil products, and we believe the current analysis delivers valid and reproducible results that support our conclusions.

Please let us know if further detail or additional analyses would be helpful.

4. Have the authors made all data underlying the findings in their manuscript fully available?

Reviewer #3: Yes

Reviewer #4: Yes

Author’s response: We appreciate the reviewers’ positive assessment regarding the availability of data underlying our findings. In accordance with the PLOS data policy, we confirm that all data supporting the conclusions of this study are fully available upon request. We are committed to transparency and reproducibility.

5. Is the manuscript presented in an intelligible fashion and written in standard English?

Reviewer #3: Yes

Reviewer #4: Yes

Author’s response: We sincerely thank the reviewers for their positive evaluation of the manuscript’s clarity and presentation. We have made every effort to ensure that the manuscript is written in clear, correct, and standard English suitable for publication.

6. Review Comments to the Author

Reviewer #3: The authors of the article made a lot of corrections, but in my opinion the work still does not meet the usual standards set for published articles. For example the authors describe methodology for Gram staining step by step; everyone knows how to do it. The same applies to the section "A microscopic method for identifying fungi"- there is a step by step recipe for how to do it.

Author’s response: We sincerely thank Reviewer #3 for the constructive feedback and for acknowledging the corrections made in the manuscript. We appreciate the valuable perspective regarding the level of detail provided in certain methodological and materials descriptions.

Our intention in including step-by-step protocols for procedures such as Gram staining and microscopic fungal identification was to ensure full transparency and reproducibility, particularly given the interdisciplinary readership of the journal. While these methods are indeed standard in many microbiology laboratories, we aimed to provide sufficient detail so that readers from diverse backgrounds, including those less familiar with routine microbiological techniques, could fully understand the processes employed in this study. However, we understand the reviewer’s suggestion and totally agree to remove it in the revised manuscript.

In the Materials section- the authors list: volumetric flasks, Petri dishes, glass rod etc. Why to do it? It is a standard equipment for every laboratory.

Author’s response: We sincerely thank you for the constructive comment and question. We included a list of common laboratory equipment to emphasize the rigor and thoroughness of the sample processing environment and to affirm that all materials conformed to standard sterile laboratory practice. However, we understand the reviewer’s suggestion that some of this detail may be unnecessarily extensive for the intended audience. Accordingly, we propose to condense these sections by removing overly detailed stepwise descriptions and standard equipment listings.

The section Data quality assurance and control methods- should be omitted.

Author’s response: We thank you dear reviewer, for the helpful suggestion regarding the "Data quality assurance and control methods" section. We agree that omitting this section will streamline the manuscript and improve its overall clarity and readability. Finally, we have removed it from the revised manuscript.

Reviewer #4: The authors on this manuscript "Microbial Quality Assessment of Niger Seed (Guizotia abyssinica (Linnaeus f.) Cassini) Oil in Gondar City: A Laboratory-Based Cross-Sectional Study," which apparently appraised or revealed Ethiopia's lack of stringent quality control and regulatory oversight, further raising concerns about public health and safety.

They further opined, from their result, that the oil processing, production, handling, and storage systems seemed not to have proper hygienic handling practices.

On careful scrutiny, I could see that the previous reviewers' queries had been addressed, although no table should be allowed to crisscross pages (take note of Table 1, pp. 11-12, and Table 3, pp. 12-13).

Summarily, if the editors are satisfied with the response to the reviewers queries, decision could be made.

Author’s response: We sincerely thank Reviewer #4 for the thorough and positive evaluation of our manuscript and for acknowledging that the reviewers’ queries have been appropriately addressed. We appreciate the insightful summary regarding the public health implications of our findings related to the hygiene practices in Niger seed oil processing and handling.

We also appreciate the constructive comment regarding the formatting of tables. We will carefully revise Tables 1 and 3 to ensure that they are presented fully on a single page, avoiding any breaks across pages, to enhance the manuscript’s readability and presentation quality. Thank you again for your valuable feedback and support for the consideration of our manuscript.

Additional changes to the revised manuscript

1. We have included the strengths and limitations of the study.

2. We have outlined the implications of the study.

3. We have also incorporated a map of the study area into the manuscript.

---

## [Editor Report · Decision Letter 3]

19 Aug 2025

Microbial Quality Assessment of Niger Seed (Guizotia abyssinica (Linnaeus f.) Cassini) Oil in Gondar City: A Laboratory-Based Cross-Sectional Study

PONE-D-24-29805R3

Dear Dr. Yohannes,

We’re pleased to inform you that your manuscript has been judged scientifically suitable for publication and will be formally accepted for publication once it meets all outstanding technical requirements.

Kind regards,

Charles Odilichukwu R. Okpala, PhD

Academic Editor

PLOS ONE

Additional Editor Comments (optional):

Thank you authors for your diligent efforts. Very acceptable for publication.
---

## [Editor Report · Acceptance letter]

PONE-D-24-29805R3

PLOS ONE

Dear Dr. Yohannes,

I'm pleased to inform you that your manuscript has been deemed suitable for publication in PLOS ONE. Congratulations! Your manuscript is now being handed over to our production team.

Kind regards,

on behalf of

Dr. Charles Odilichukwu R. Okpala

Academic Editor

PLOS ONE